# UniCtrl: Improving the Spatiotemporal Consistency of Text-to-Video Diffusion Models via Training-Free Unified Attention Control

**Tian Xia** *                                                          *tianxia@umich.edu*
*University of Michigan*

**Xuweiyi Chen** *‡                                                   *xuweic@email.virginia.edu*
*University of Virginia*

**Sihan Xu** †‡                                                       *sihanxu@umich.edu*
*University of Michigan*

**Reviewed on OpenReview:** *https://openreview.net/forum?id=x2uFJ79OjK*

## Abstract

Video Diffusion Models have been developed for video generation, usually integrating text and image conditioning to enhance control over the generated content. Despite the progress, ensuring consistency across frames remains a challenge, particularly when using text prompts as control conditions. To address this problem, we introduce **UniCtrl**, a novel, plug-and-play method that is universally applicable to improve the spatiotemporal consistency and motion diversity of videos generated by text-to-video models without additional training. UniCtrl ensures semantic consistency across different frames through *cross-frame self-attention control*, and meanwhile, enhances the motion quality and spatiotemporal consistency through *motion injection* and *spatiotemporal synchronization*. Our experimental results demonstrate UniCtrl's efficacy in enhancing various text-to-video models, confirming its effectiveness and universality.

## 1 Introduction

Diffusion Models (DMs) have excelled in image generation, offering enhanced stability and quality over methods like GANs (Goodfellow et al., 2020; Karras et al., 2019; 2020) and VAEs (Kingma & Welling, 2014; Van Den Oord et al., 2017; Ramesh et al., 2021). The superior image generation capability of diffusion models stems from the critical role of the attention mechanism(Rombach et al., 2022a; Peebles & Xie, 2023; Dhariwal & Nichol, 2021). Foundational studies (Sohl-Dickstein et al., 2015; Ho et al., 2020; Song et al., 2020a; 2023; Luo et al., 2023; Karras et al., 2022) established the groundwork for DMs' capabilities in efficiently scaling up with diverse data. Recent advancements (Rombach et al., 2022a; Ramesh et al., 2022; Nichol et al., 2022; Saharia et al., 2022; Xu et al., 2023b; Zhang et al., 2023b; Mou et al., 2023) have further improved controllability and user interaction, enabling the creation of images that better reflect user intentions.

Recently, Video Diffusion Models (VDMs) (Ho et al., 2022b) have been proposed for utilizing DMs for video generation tasks. VDMs are capable of generating videos with a wide variety of motions in text-to-video

---

*Authors contributed equally to this work.
†Corresponding author and project lead.
‡Work completed at PixAI.art

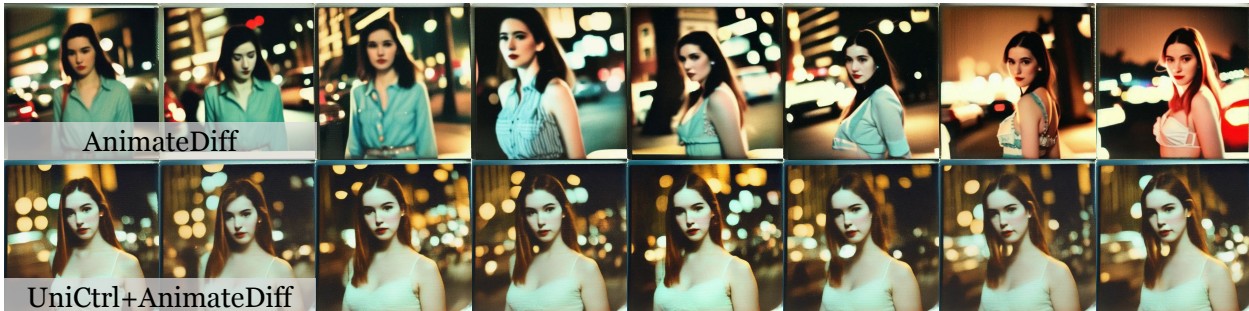

*polaroid photo, night photo, photo of 24 y.o beautiful woman, pale skin, bokeh*

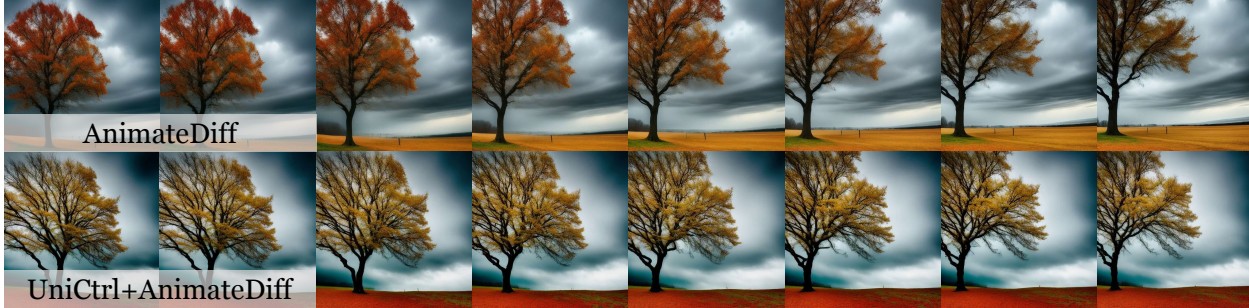

*professional movie, movie of autumn landscape, dramatic lighting, gloomy, cloud weather, tree*

Figure 1: UniCtrl for Video Generation. we propose **UniCtrl**, a concise yet effective method to significantly improve the temporal consistency of videos generated by diffusion models yet also preserve the motion. UniCtrl requires no additional training and introduces no learnable parameters, and can be treated as a plug-and-play module at inference time.

synthesis tasks, supported by the integration of text encoders (Radford et al., 2021), self attention, cross attention and temporal attention, as shown in (Guo et al., 2023b; Zhang et al., 2023a; Hu et al., 2022; Ho et al., 2022a; Blattmann et al., 2023a;b). Many open-source text-to-video models have been introduced, including ModelScope(Wang et al., 2023), AnimateDiff(Guo et al., 2023b), VideoCrafter(Chen et al., 2023) and so on. These models typically require a pre-trained image generation model, e.g., Stable Diffusion(SD) (Rombach et al., 2022b), and introduce additional temporal or motion modules. However, unlike images that contain rich semantic information, text conditions are more difficult to ensure consistency between different frames of the generated video. At the same time, some work also uses image conditions to achieve image-to-video generation with improved spatial semantic consistency (Guo et al., 2023a; Hu & Xu, 2023; Blattmann et al., 2023a). Some works have proposed the paradigm of text-to-image-to-video(Girdhar et al., 2023), but image conditions alone cannot effectively control the motion of videos. Combining text and image conditions leads to enhanced spatiotemporal consistency in a text-and-image-to-video workflow (Zhang et al., 2023c; Chen et al., 2023; 2024; Xing et al., 2023; Gu et al., 2023b), but these methods require additional training.

To this end, our research goal in this work is to develop an effective plug-and-play method that is training-free, and can be applied to various text-to-video models to improve the performance of generated videos. To solve this problem, we first attempt to ensure that the semantic information between each frame of the video is consistent in principle. As the attention mechanism plays a significant role, this principle draws inspiration from previous research in attention-based control (Hertz et al., 2023; Tumanyan et al., 2023; Cao et al., 2023; Xu et al., 2023a). These works have demonstrated in DMs that the queries in attention layers determine the spatial information of the generated image, and correspondingly, values determine the semantic information. We observe that this finding also holds in VDMs and propose the *cross-frame self-attention control* method. We thus apply the keys and values of the first frame in self-attention layers to each frame and achieve satisfying consistency in the generated video.

Secondly, we observe that as the video's consistency improves, the motions within videos tend to become less pronounced. To solve this problem, we propose the *motion injection* mechanism. Based on the assumption that queries control spatial information (Cao et al., 2023), we divide the sampling process into two branches:

an output branch for *cross-frame self-attention control* and a motion branch without any attention control. We reserve the queries in the motion branch as motion queries, and use the corresponding motion queries in the output branch. Through the *cross-frame self-attention control* and *motion injection*, we ensure that the semantic information between each video frame is consistent, while the motion is preserved.

Lastly, we note that motion queries cannot guarantee the spatiotemporal consistency of video. Observing that the output of the output branch has a better spatiotemporal consistency, we further propose *spatiotemporal synchronization*, that is, before each sampling step, the latent of the output branch is copied as the initial value of the latent of the motion branch. Our UniCtrl framework combines the above three methods into a plug-in-and-play method that can improve the quality of spatiotemporal consistency and motion quality of the generated videos, while ensuring the consistency of the semantic information of each frame of the video. A recent survey (Melnik et al., 2024) also emphasizes the importance of enhancing spatiotemporal coherence and motion consistency to advance video diffusion models. In the other domains related to video generation, various methods attempt to address similar challenges (Huang et al., 2023; Yang et al., 2023). Through experiments, several text-to-video models have been improved after applying the UniCtrl method, proving the effectiveness and universality of UniCtrl. As illustrated in Figure 1, UniCtrl plays a significant role in improving spatiotemporal consistency and preserving the motion dynamics of generated frames. This method can be readily applied during inference without the need for parameter tuning.

## 2 Related Work

**Video Generation**   Many previous efforts have explored the task of video generation, e.g., GAN-based models (Skorokhodov et al., 2022; Tian et al., 2021; Brooks et al., 2022) and transformer-based models(Hong et al., 2022; Villegas et al., 2022; Wu et al., 2021; 2022). Recently, following that Diffusion models (DMs) (Sohl-Dickstein et al., 2015; Ho et al., 2020; Song et al., 2020a; 2023; Luo et al., 2023; Karras et al., 2022) have achieved remarkable results in image generation (Rombach et al., 2022a; Ramesh et al., 2022; Nichol et al., 2022; Saharia et al., 2022; Nichol & Dhariwal, 2021), video diffusion models (VDMs) (Ho et al., 2022b) has also demonstrated their capabilities in video generation (Blattmann et al., 2023b; Girdhar et al., 2023; Gu et al., 2023b; Ho et al., 2022a; Hu et al., 2022; Singer et al., 2022; Zhang et al., 2023a; Guo et al., 2023b; Chen et al., 2023; 2024; Xing et al., 2023; Wang et al., 2023). At present, VDMs are mainly implemented by adding additional temporary layers to 2D UNet, which leads to a lack of cross-frame constraints in the training process of the 2D UNet model. Some methods (Wu et al., 2023; Qiu et al., 2023; Gu et al., 2023a) tried to use a training-free method to make the generated videos more smooth. However, how to maintain the cross-frame consistency in videos generated by VDMs remains unresolved.

**Attention Control in Diffusion Models**   Different from models that require training(Zhang et al., 2023b; Xu et al., 2023b; Mou et al., 2023), attention control (Hertz et al., 2023; Tumanyan et al., 2023; Cao et al., 2023; Xu et al., 2023a; Ge et al., 2023b) is a training-free method which has been widely applied in the task of image editing. Previous work has found that Attention Control can be used to ensure both semantic (Cao et al., 2023) and spatial consistency (Hertz et al., 2023; Tumanyan et al., 2023; Ge et al., 2023b) in image editing. InfEdit(Xu et al., 2023a) unified the control of semantic consistency and spatial consistency for the first time, proposing unified attention control (UAC). Text2Video-Zero (Khachatryan et al., 2023) applies frame-level self-attention on text-to-image synthesis methods and enables text-to-video through manipulating motion dynamics in latent codes. Some work has introduced attention control to VDMs for video editing (Liu et al., 2023; Geyer et al., 2023; Khandelwal, 2023), but no one has improved the consistency of generated videos through video diffusion by attention control. Is that possible to introduce UAC into VDMs to ensure cross-frame semantic consistency and spatial consistency throughout the video is an interesting and worthwhile question to explore.

## 3 Preliminary

### 3.1 Diffusion Models (DMs)

Diffusion Models (DMs) (Song et al., 2020a; Ho et al., 2020; Song et al., 2023; Karras et al., 2022) are a type of generative model trained via score matching (Hyvärinen & Dayan, 2005; Lyu, 2012; Song & Ermon,

2019; Song et al., 2020b). The forward process gradually adds noise to data to make it follow the Gaussian distribution: $z_t = \mathcal{N}(z_t; \sqrt{\bar{a}_t}z_0, (1 - \sqrt{\bar{a}_t})I)$, where $z_0$ are samples from the data distribution and $\alpha_1, \ldots, \alpha_T$ are from a variance schedule. In Ho et al. (2020), this process is re-parameterized into the following form:

$$z_t = \sqrt{\bar{\alpha}_t}z_0 + \sqrt{1 - \bar{\alpha}_t}\varepsilon, \varepsilon \sim \mathcal{N}(0, I) \tag{1}$$

The training objective of DMs is predict the added noise via a neural network $\varepsilon_\theta$, to reconstruct the original input from the noisy sample by minimizing the distance $d(\cdot, \cdot)$:

$$\min_\theta \mathbb{E}_{z_0, \varepsilon, t}\big[d\left(\varepsilon - \varepsilon_\theta(z_t, t)\right)\big] \tag{2}$$

The sampling process of DMs (Ho et al., 2020; Song et al., 2020a) is iterative and can be represented in the form with different noise schedule $\sigma_t$:

$$
\begin{aligned}
z_{t-1} = &\sqrt{\bar{\alpha}_{t-1}}\left(\frac{z_t - \sqrt{1 - \bar{\alpha}_t}\varepsilon_\theta(z_t, t)}{\sqrt{\bar{\alpha}_t}}\right) \quad \text{(predicted } z_0\text{)} \\
&+ \sqrt{1 - \bar{\alpha}_{t-1} - \sigma_t^2} \cdot \varepsilon_\theta(z_t, t) \qquad \text{(direction to } z_t\text{)} \\
&+ \sigma_t\varepsilon \quad \text{where } \varepsilon \sim \mathcal{N}(0, I) \qquad \text{(random noise)}
\end{aligned}
\tag{3}
$$

Latent Diffusion Models (LDMs) (Rombach et al., 2022a) encodes samples into latents using an encoder $\mathcal{E}$, such that $z_0 = \mathcal{E}(x_0)$. Also, the output is reconstructed via a decoder $\mathcal{D}$, represented by $\mathcal{D}(z)$. This approach has led to improvements in stability and efficiency in the training and generation process.

Video Diffusion Models(VDMs) (Ho et al., 2022b) extended the application of DMs to the domain of video generation, adapting the framework to handle 4D video tensors in the form of $frames \times height \times width \times channels$, which we can use $z_t^f$ to describe the frame $f + 1$ at the timestep $t$.

### 3.2 Attention Control

We follow the notation from (Hertz et al., 2023; Xu et al., 2023a). In the fundamental unit of the diffusion UNet model, there are two main components: cross-attention and self-attention blocks. The process begins with the linear projection of spatial features to generate queries ($Q$). In the self-attention block, both keys ($K$) and values ($V$) are derived from the spatial features through linear projection. Conversely, for the cross-attention part, text features undergo a linear transformation to form keys ($K$) and values ($V$). The attention mechanism (Vaswani et al., 2017) can be described as:

$$\text{ATTENTION}(Q, K, V) = MV = \text{softmax}\left(\frac{QK^T}{\sqrt{d}}\right)V \tag{4}$$

Mutual Self-Attenion Control (MasaCtrl) is proposed by (Cao et al., 2023), and find that replacing the $Q$s in attention layers while keeping the $K$s and $V$s same, can change the spatial information of generated images but keeping the semantic information preserved. This technique can help diffusion in spatial-level editing, e.g. a sitting dog to a running dog. Here we use $(\cdot)^{\text{src}}$ to represent the tensor obtained from the source image and $(\cdot)^{\text{tgt}}$ for the target output we want, we can use the following formula to define the MasaCtrl algorithm.

$$out = \text{ATTENTION}(Q^{\text{tgt}}, K^{\text{src}}, V^{\text{src}})$$

Cross-Attenion Control (P2P) is a method mentioned in (Hertz et al., 2023), which is for semantic-level image editing (e.g. dog to cat). This work observed that different $V$s from different text prompts decide the semantic information of generated images. If attention maps ($M$) in cross-attention layers have been reserved, but use different Vs for the attention calculation, most of the spatial information will be preserved. We here use $(\cdot)^{\text{src}}$ to represent the tensor obtained from the source image and prompt and $(\cdot)^{\text{tgt}}$ for the target output with target prompt, this algorithm can be described as following as same $Q^{\text{src}}, K^{\text{src}}$ lead to same $M^{\text{src}}$:

$$out = \text{CROSSATTENTION}(Q^{\text{src}}, K^{\text{src}}, V^{\text{tgt}})$$

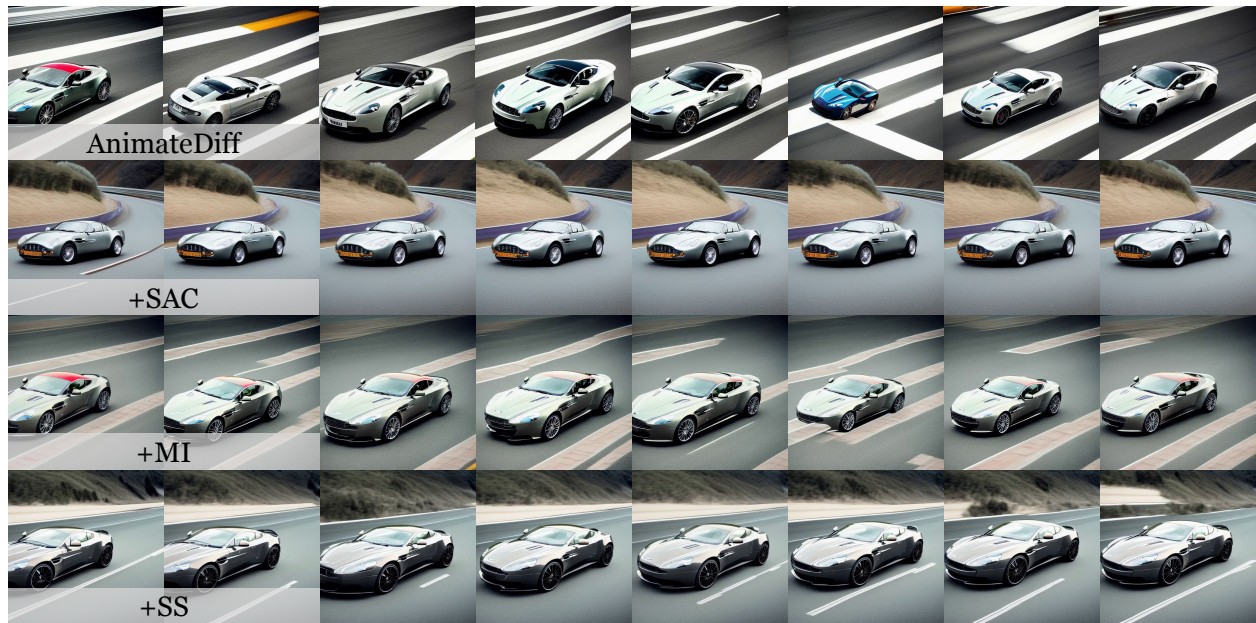

*Aston Martin is moving*

Figure 2: The first row demonstrates the original frames generated with the baseline model, in which the vehicle and the road are inconsistent across the frames. The second row shows frames generated with baseline model augmented with *cross-frame **S**elf-**A**ttention **C**ontrol* (SAC). While it maintains incredible spatiotemporal consistency, it exhibits little motion. The third row explains frames augmented with SAC and ***M**otion **I**njection* (MI). Although MI injects more motion in addition to SAC, the results demonstrate that it falls short on spatiotemporal consistency again. The fourth row contains frames further augmented with ***S**patiotemporal **S**ynchronization* (SS) in addition to SAC and MI, which improves spatiotemporal consistency over the results from the third row and achieves a balance between motion and spatiotemporal consistency, both in-frame and cross-frame.

## 4 UniCtrl: Cross-Frame Unified Attention Control

In the text-to-video task, it is difficult to ensure the consistency between different frames of the generated video due to the lack of semantic level conditions and the constraint of different frames in the 2D UNet layers. Based on the previous work of DMs(Hertz et al., 2023; Tumanyan et al., 2023; Xu et al., 2023a; Cao et al., 2023), it is found that the queries in the attention layers determine the spatial information of the generated images, and correspondingly, the values determine the semantic information. We assume these properties still exist in VDMs.

We first analyzed the role of keys and values in the self-attention layers of VDMs, and then analyzed the role of queries in all the attention layers. Inspired by InfEdit (Xu et al., 2023a), we then propose the cross-frame unified attention control to achieve both semantic level consistency and better spatiotemporal consistency. Lastly, we apply *Spatiotemporal Synchronization* (SS) by replacing the latent of the motion branch with the latent from the output branch at each sampling step. We demonstate the effectiveness of each module of UniCtrl in Figure 2.

### 4.1 Cross-Frame Self-Attention Control

Previous work (Cao et al., 2023; Hertz et al., 2023; Tumanyan et al., 2023; Xu et al., 2023a) has observed that queries in the attention mechanism form the layout and semantic information of generated images(Cao et al., 2023; Xu et al., 2023a; Tumanyan et al., 2023), while values contribute to the semantic information(Hertz et al., 2023; Xu et al., 2023a). Therefore, we hypothesize that using the same values in the attention of different frames can ensure cross-frame consistency. Additionally, we observed that the mismatch between keys and values will degrade the quality of generated videos through our experiment and we provide one

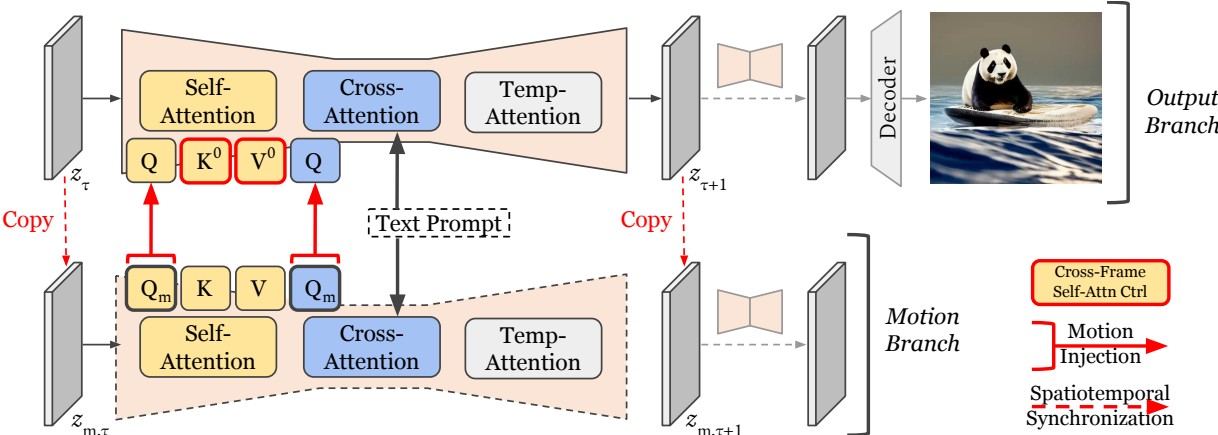

Figure 3: In our framework, we use key and value from the first frame as represented by $K^0$ and $V^0$ in the self-attention block. We also use another branch to keep the motion query $Q_{\mathrm{m}}$ for motion control. At the beginning of every sampling step, we let the motion latent equal to the sampling latent, to avoid spatial-temporal inconsistency. Note that in the actual workflow, Q replacement occurs only in cross-attention, as the Q in the self-attention blocks of both branches are always the same. We explain details of our framework in Algorithm 1 and Algorithm 2.

qualitative example in Section 5.4. Consequently, in our *cross-frame **S**elf-**A**ttention **C**ontrol* (SAC), we inject both keys and values from the first frame, as detailed in Algorithm 1, to every other frame at each self-attention layer during denoising. We showcase one example of the effectiveness of SAC by comparing the first row and the second row in Figure 2.

### 4.2 Motion Injection

In our experiments, detailed in Section 5, we identify a significant limitation associated with *cross-frame self-attention control*: it tends to produce overly similar consecutive frames, leading to minimal motion within the video sequence. Again, as described in (Cao et al., 2023; Xu et al., 2023a), the queries in self-attention and cross-attention determine the video's spatial information, which corresponds to motion. Therefore, by preserving the original queries, we can maintain the motion effect.

---

**Algorithm 1** Cross-Frame Self-Attention Control

1: **Input:** Hidden State $z$
2: $bs, c, f, h, w = z.\text{shape}()$
3: $z^0 = z[:, :, 0, :, :].\text{unsqueeze}(2)$ # Get the 1st frame.
4: $z^0 = z^0.\text{repeat}(1, 1, f, 1, 1)$
5: $Q = \text{to\_q}(z)$
6: $K^0 = \text{to\_k}(z^0)$ # Get the Key of the first frame.
7: $V^0 = \text{to\_v}(z^0)$ # Get the Value of the first frame.
8: $out = \text{SELFATTENTION}(Q, K^0, V^0)$
9: **Output:** $out$

---

As in Figure 3, we have divided the inference process into two branches: the output branch, which undergoes *cross-frame self-attention control*, and the motion branch, which does not involve attention control. We retain the queries in the motion branch as motion queries and use the corresponding motion queries in the output branch. Here, we denote motion queries as $Q_{\mathrm{m}}$. The method for *motion injection* can be expressed by the following formula: $out = \text{ATTENTION}(Q_{\mathrm{m}}, \cdot, \cdot)$, where ATTENTION refers to both of self- and cross-attention layers. This method is designed to fully retain the motion of the original video. However, in practice, a trade-off between motion preservation and spatiotemporal coherence remains evident. Consequently, we propose an additional refinement: selectively preserving motion at specific steps throughout the sampling process to enhance control. To this end, we introduce a technique that modulates motion through the integration of a *motion injection* degree, $c$, which is defined within the interval $0 \leq c \leq 1$. This approach is further detailed and formalized presented in Algorithm 2 and the following outlines this approach in detail :

$$out = \begin{cases} \text{ATTENTION}(Q_{\mathrm{m}}, \cdot, \cdot) & t \geq (1-c) \times T \\ \text{ATTENTION}(Q, \cdot, \cdot) & t < (1-c) \times T \end{cases}$$

### 4.3 Spatiotemporal Synchronization

Upon closer inspection of the results, notably exemplified by the third row of Figure 2, it becomes evident that simply injecting spatial information from the original video is insufficient for guaranteeing spatiotemporal consistency. Considering that output branch yields more spatiotemporally consistent results, we further propose ***S**patiotemporal **S**ynchronization* (SS): the latent of the output branch is copied as the initial value of the latent for the motion branch before each sampling step. By doing so, our method can simultaneously ensure the semantic consistency of the generated video and improve the quality of spatiotemporal consistency and preserve the degree of motion diversity. We present a qualitative example to demonstrate the effectiveness of SS by comparing the results depicted in the third and fourth rows of Figure 2.

---

**Algorithm 2** Motion Injection

1: **Input:**
 Video Diffusion Model **VDM**
 Sequence of timesteps $\tau_0 > \tau_1 > \cdots > \tau_N$
 Text Condition $c$
 Timestep condition for Motion Control $t$
2: $z_{\mathrm{m},\tau_0} = z_{\tau_0} \sim \mathcal{N}(\mathbf{0}, \boldsymbol{I})$
3: **for** $n = 0$ to $N - 1$ **do**
4:  $z_{\mathrm{m},\tau_n} = z_{\tau_n}$
5:  $Q_{\mathrm{m}} \leftarrow \mathbf{VDM}(z_{\mathrm{m},\tau_n}, c, \tau_n)$
6:  **if** $t \geq \tau_n$ **then**
7:   $z_{\tau_{n+1}} = \mathbf{VDM}(z_{\tau_n}, c, \tau_n)\{Q \leftarrow Q_{\mathrm{m}}\}$
8:  **else**
9:   $z_{\tau_{n+1}} = \mathbf{VDM}(z_{\tau_n}, c, \tau_n)$
10:  **end if**
11: **end for**
12: **Output:** $z_{\tau_N}$

---

### 4.4 Cross-Frame Unified Attention Control

As illustrated in Figure 3, we integrate *cross-frame self-attention control*, *motion injection* and *spatiotemporal synchronization* into a cohesive framework termed ***C**ross-**F**rame **U**nified **A**ttention **C**ontrol*(UniCtrl). In the output branch, we employ the key $K^0$ and value $V^0$ derived from the initial frame to maintain cross-frame semantic consistency. A separate branch is utilized to preserve the query specifically for motion control, replacing the output branch's query with $Q_{\mathrm{m}}$ from the motion control branch. To prevent spatiotemporal inconsistencies, we synchronize the latent representation of the motion branch with the output branch's preceding latent state before each sampling step.

## 5 Experiments

In this section, we evaluate the effectiveness of UniCtrl. We discuss metrics, backbones and baseline in Section 5.1. Then we include both qualitative comparisons in Section 5.2 and quantitative comparisons in Section 5.3 to showcase the effectiveness of UniCtrl in terms spatiotemporal consistency and motion preservation. Additionally, we explore the contribution of each component within UniCtrl, the *motion injection* degree, and the specific design choice of swapping Key and Value together in the SAC procedure, as detailed in Section 5.4.

### 5.1 Experimental Setup

To evaluate the effectiveness of our model, we collect prompts from two datasets UCF-101 (Soomro et al., 2012) and MSR-VTT (Xu et al., 2016) for generating videos. Following Ge et al(Ge et al., 2023a), we use the same UCF-101 prompts for our experiments. We also randomly selected 100 unique prompts from the MSR-VTT dataset for our evaluation. Those two parts of data consist of our dataset for evaluation. To mitigate the stochasticity inherent in diffusion models, all experiments were conducted using multiple random seeds. The scores reported in the tables represent the average results across these runs, and we provide the corresponding standard deviations in the Appendix. Next, we briefly introduce evaluation metrics and we also provide details in the Appendex.

**Metric**  To quantitatively evaluate our results, we consider standard metrics following (Singer et al., 2022; Wu et al., 2023):

• *DINO*: To evaluate the spatiotemporal consistency in the generated video, we employ DINO(Oquab et al., 2024) to compare the cosine similarity between the initial frame and subsequent frames. In our experiments, we utilize the DINO-vits16(Caron et al., 2021) model to compute the DINO cosine similarity.

- *RAFT*: To compare the magnitude of motion in the videos, we utilize RAFT(Teed & Deng, 2020) to estimate the optical flow, thereby inferring the degree of motion. We utilize the off-the-shelf RAFT model from torchvision (maintainers & contributors, 2016).

- *FVD*: To assess video quality, particularly the quality of individual frames during realistic motions, we employ the Content-Debiased FVD metric (Ge et al., 2024) to evaluate the generated videos. In our experiments, we use the official library to compute the FVD.

- *FVMD*: To evaluate the quality of motion consistency in video generation, we use the Fréchet Video Motion Distance (FVMD) metric (Liu et al., 2024) to assess temporal coherence based on velocity and acceleration patterns. In our experiments, we employ the official library to calculate the FVMD with the ground truth videos from our prompt dataset.

**Backbones**  Since our method is plug-and-play, we decide to evaluate our methods on a few popular baselines:

- *AnimateDiff* (Guo et al., 2023b) introduces a practical framework for adding motion dynamics to personalized text-to-image models, such as those created by Stable Diffusion, without the need for model-specific adjustments. Central to AnimateDiff is a motion module, trainable once and universally applicable across personalized text-to-image models derived from the same base model, leveraging transferable motion priors from real-world videos for animation.

- *VideoCrafter* (Chen et al., 2023) introduces two novel diffusion models for video generation: T2V, which synthesizes high-quality videos from text inputs, achieving cinematic-quality resolutions up to 1024×576, and I2V, the first open-source model that transforms images into videos while preserving content, structure, and style. Both models represent significant advancements in open-source video generation technology, offering new tools for researchers and engineers. We use the T2V version in our experiments.

- *AnimateLCM* (Wang et al., 2024) introduces a novel framework for efficient and high-quality video generation by leveraging consistency learning. It builds upon the principles of the Consistency Model (CM) and Latent Consistency Model (LCM) from image diffusion models, accelerating video generation with minimal steps. AnimateLCM employs a decoupled consistency learning strategy, separating the learning of image generation priors and motion generation priors.

**Baseline**  We select FreeInit (Wu et al., 2023) as our baseline since FreeInit is a training-free method that attempts to improve the subject appearance and temporal consistency of generated videos through iteratively refining the spatial-temporal low-frequency components of the initial latent during inference. However, given that our method UniCtrl operates on the attention mechanism and FreeInit adjusts the frequency domain of the latent space, UniCtrl and FreeInit are orthogonal approaches. Both are training-free methods capable of enhancing the spatiotemporal consistency of generated videos via diffusion models. We demonstrate the integration of UniCtrl and FreeInit both in 5.2 and 5.3.

## 5.2  Qualitative Comparisons

Qualitative comparisons, depicted in Figure 4, reveal that our UniCtrl method markedly improves spatiotemporal consistency and maintains motion diversity. For example, with the text prompt "walking with a dog", FreeInit produces inconsistent appearances for both the lady and the dog, whereas UniCtrl ensures consistent representations of both entities. Furthermore, when processing the prompt "A young woman walks through flashing lights", UniCtrl maintains the detailed features of the young woman's dress while ensuring her walking motion remains natural, in contrast to the vanilla AnimateDiff model. Additionally, we demonstrate UniCtrl's flawless integration with FreeInit, consistently preserving the young woman's appearance.

We show additional qualitative results in Figure 5 and Figure 6 to demonstrate the efficacy of UniCtrl to significantly improving spatiotemporal consistency and preserve motion dynamics for a diverse set of prompts.

## 5.3  Quantitative Comparisons

For quantitative comparisons, the quantitative results on UCF-101 and MSR-VTT are reported in Table 1. We compare the backbones and backbones augmented by UniCtrl and FreeInit respectively. According to

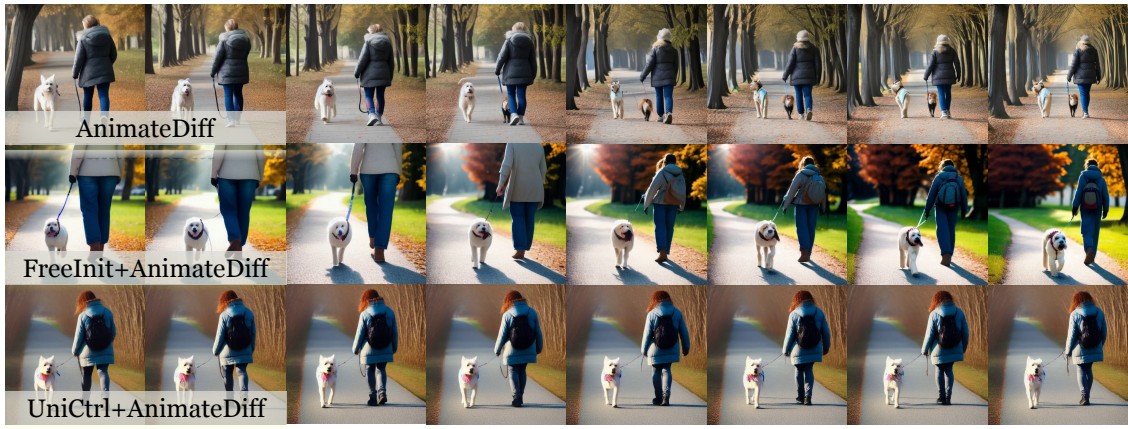

*walking with dog*

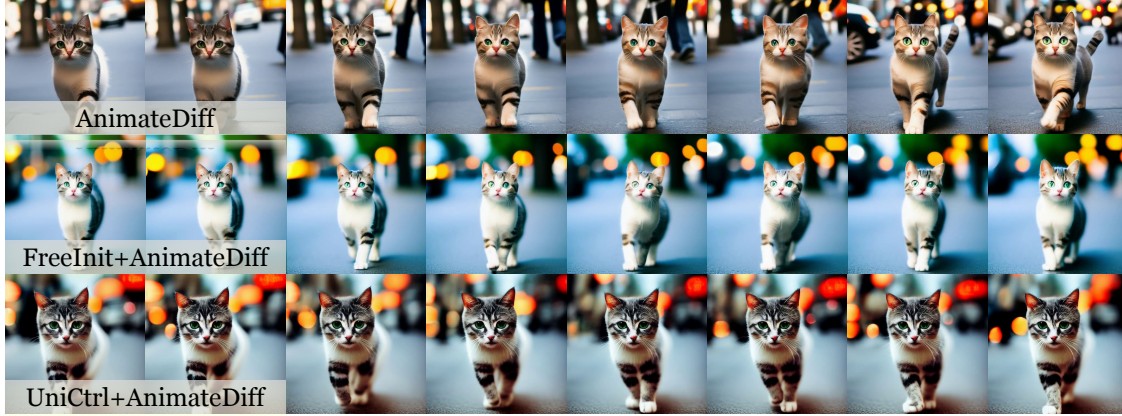

*cute cat walking on the city streets*

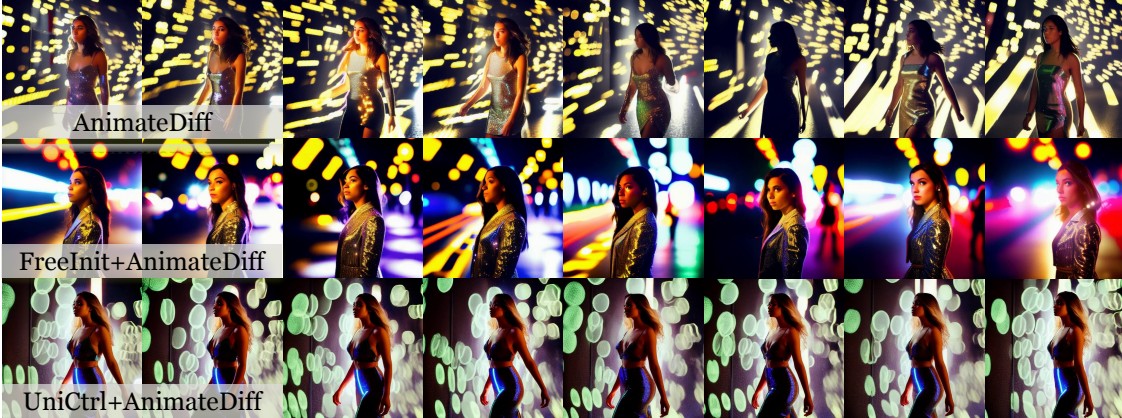

*A young woman walks through flashing lights*

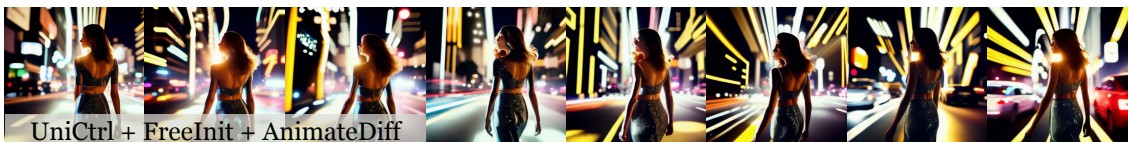

Figure 4: **Qualitative Comparisons**. We demonstrate UniCtrl's adaptability to diverse prompts, enhancing temporal consistency and preserving motion diversity. Comparative inference results with FreeInit are presented for context. Additionally, we demonstrate UniCtrl's seamless integration with FreeInit.

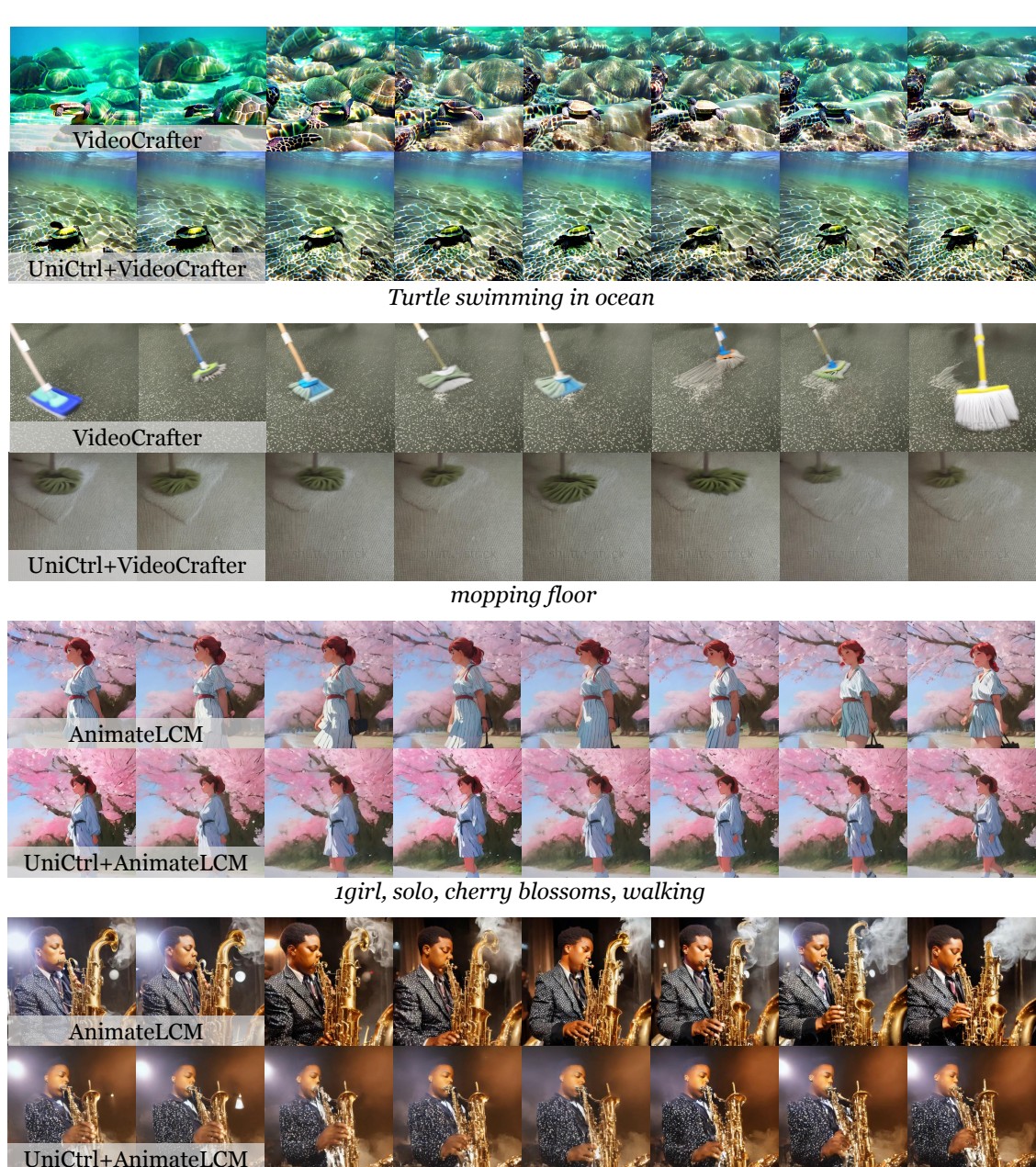

Figure 5: We provide additional qualitative examples across various backbones to demonstrate UniCtrl's capability in enhancing spatiotemporal consistency while effectively preserving motion dynamics.

the metrics, UniCtrl significantly improves the spatiotemporal consistency in the generated videos across all backbones on both prompt sets from 2.12 to 2.44. While FreeInit achieves remarkable improvements over spatiotemporal consistency, we found that UniCtrl outperforms FreeInit on the strength of motions on both AnimateDiff and VideoCrafter by a large margin from 11.67 to 17.65. Note that we purposely chose the *motion injection* degree $c = 1$ to demonstrate how UniCtrl can preserve the motion compared with FreeInit. However, there still exists a trade-off between spatiotemporal consistency and motion diversity. In section 5.4, we show spatiotemporal consistency can be further improved with a smaller *motion injection* degree. Thus, we recommend *motion injection* degree $c = 0.4$ for real-world applications. Lastly, we showcase how UniCtrl and FreeInit can improve AnimateDiff together and we found this integration can improve spatiotemporal consistency together. We will introduce the details of how we integrate UniCtrl and FreeInit

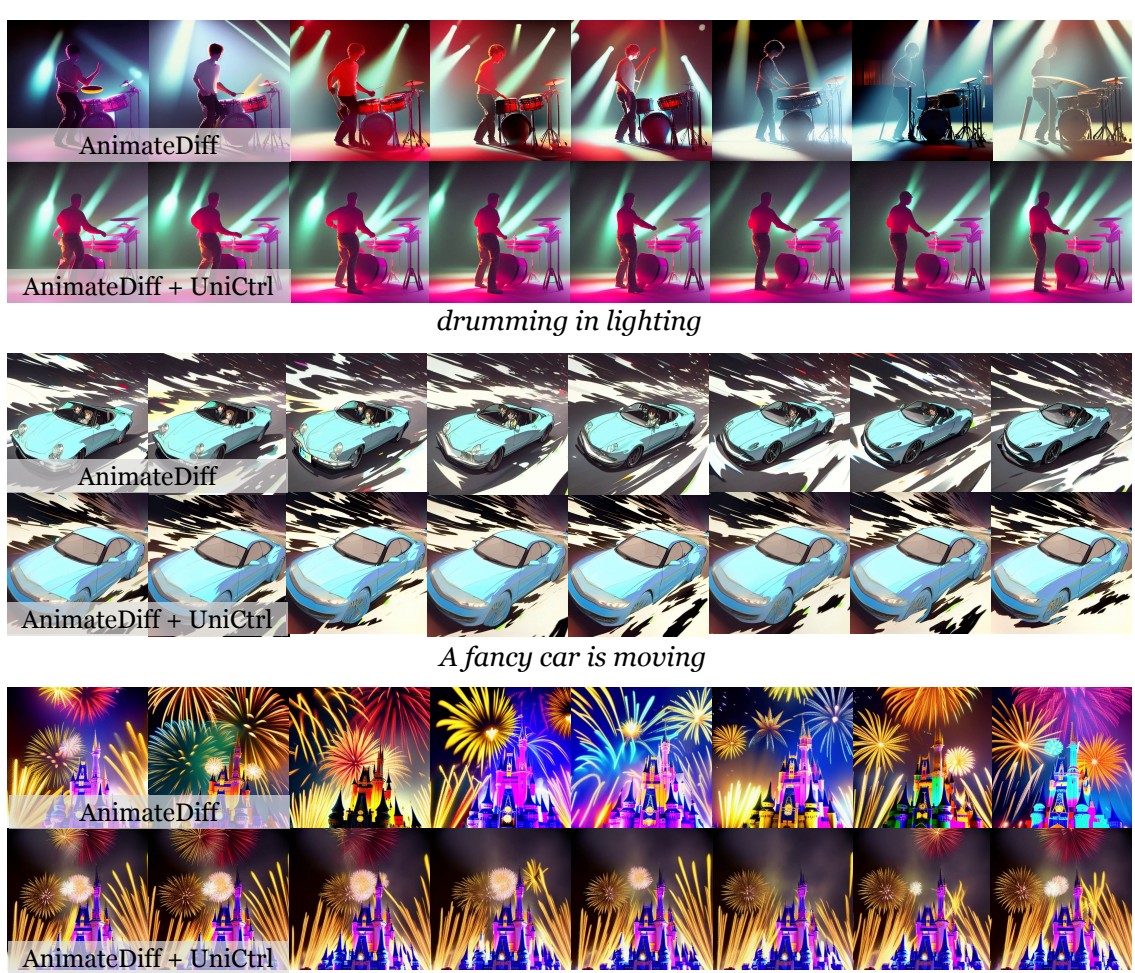

Figure 6: We present more qualitative examples to show UniCtrl's ability to improve spatiotemporal consistency and significantly preserve motion dynamics.

Table 1: Quantitative Comparisons on UCF-101 and MSR-VTT. UniCtrl significantly improves the temporal consistency while keeping the motion in the generated videos. $c$ indicates the *motion injection* degree. $I$ indicates the number of iterations for FreeInit.

| Method | DINO (↑) | RAFT (↑) |
|---|---|---|
| AnimateDiff (Guo et al., 2023b) | 94.26 | 25.44 |
| FreeInit + AnimateDiff ($I = 3$) | 96.04 | 11.62 |
| UniCtrl + AnimateDiff ($c = 1$) | 96.38 (+00.34) | 23.29 (+11.67) |
| VideoCrafter (Chen et al., 2023) | 93.53 | 29.20 |
| FreeInit + VideoCrafter ($I = 3$) | 96.75 | 7.46 |
| UniCtrl + VideoCrafter ($c = 1$) | 95.55 (−01.20) | 25.11 (+17.65) |
| AnimateLCM (Wang et al., 2024) | 92.96 | 20.80 |
| FreeInit + AnimateLCM ($I = 3$) | 94.90 | 14.64 |
| UniCtrl + AnimateLCM ($c = 1$) | 95.40 (+00.50) | 17.53 (+2.89) |
| UniCtrl + FreeInit + AnimateDiff ($I = 3, c = 1$) | 96.85 | 9.82 |

in the Appendix. Note that we obtained different scores for FreeInit because we randomly sampled a different set of prompts from MSR-VTT and we used our own implementations for FreeInit on VideoCrafter since we cannot find official implementation.

Furthermore, regarding video quality and motion consistency, we present the quality results on UCF-101 in Table 2. We compare AnimateDiff with FreeInit augmented and Unictrl augmented. The metrics show that UniCtrl significantly enhances the quality of individual video frames, improving the score from 1069.90 to 819.74. Additionally, we observe a substantial improvement

Table 2: Quality results on UCF-101, which show a significant improvement in both video quality and motion consistency in video generation by UniCtrl.

| Method | FVD ($\downarrow$) | FVMD ($\downarrow$) |
|---|---|---|
| AnimateDiff (Guo et al., 2023b) | 1069.90 | 27124.17 |
| FreeInit + AnimateDiff | 958.97 | 25078.05 |
| UniCtrl + AnimateDiff | 819.74 | 8864.07 |

in FVMD, decreasing from 27,124.17 to 8,864.07, which indicates a marked enhancement in motion consistency. These results further demonstrate our model's capability to produce higher-quality videos with improved motion coherence, highlighting the significant effectiveness of our approach in advancing text-to-video models.

### 5.4 Ablation Study

In this section, we assess the impact of each UniCtrl module and the efficacy of the *motion injection* degree. We further corroborate our design choice of swapping Key and Value together in the SAC procedure through the subsequent ablation studies.

**The Impact of SAC, MI, and SS** To assess the contribution of the SAC, we conducted experiments on both datasets using AnimateDiff as the baseline, incorporating our pipeline but disabling SAC. For *motion injection*, we set the *motion injection* degree to be 1. The findings reveal our method without SAC works the same on the backbone because the motion branch is exactly the same as the output branch. This observation underscores the critical role of SAC in enhancing spatiotemporal consistency, as evidenced by the comparisons with the scores from the vanilla AnimateDiff.

In exploring the significance of *Motion Injection* (MI), additional tests were performed on both datasets with AnimateDiff serving as the baseline, this time with MI deactivated. The results

Table 3: Ablation results on UCF-101 and MSR-VTT. We ablate each module of UniCtrl and show the effectiveness of them each. $c$ indicates the *motion injection* degree.

| Method | DINO ($\uparrow$) | RAFT ($\uparrow$) |
|---|---|---|
| AnimateDiff (Guo et al., 2023b) | 94.26 | 25.44 |
| UniCtrl w/o SAC + AnimateDiff | 94.26 | 25.42 |
| UniCtrl w/o MI + AnimateDiff | 98.08 | 4.12 |
| UniCtrl w/o SS + AnimateDiff | 94.26 | 21.90 |
| only SAC + AnimateDiff | 98.08 | 4.12 |
| only MI + AnimateDiff | 94.26 | 25.42 |
| only SS + AnimateDiff | 94.26 | 25.42 |
| UniCtrl ($c = 0$) + AnimateDiff | 98.08 | 4.12 |
| UniCtrl ($c = 0.2$) + AnimateDiff | 97.41 | 9.04 |
| UniCtrl ($c = 0.4$) + AnimateDiff | 96.69 | 15.56 |
| UniCtrl ($c = 0.6$) + AnimateDiff | 96.46 | 20.16 |
| UniCtrl ($c = 0.8$) + AnimateDiff | 96.37 | 22.50 |
| UniCtrl ($c = 1.0$) + AnimateDiff | 96.38 | 23.29 |

indicated a notable consistency with the baseline, yet with a substantial reduction in motion diversity. This was quantitatively supported by a decrease in the RAFT score from 31.81 to 4.19. Such a marked disparity highlights MI's vital contribution to maintaining the motion dynamics.

Finally, the necessity of the *Spatiotemporal Synchronization* (SS) was examined by excluding SS from our pipeline and conducting experiments across both datasets, again using AnimateDiff as the reference point and setting *motion injection* degree to be 1. The outcomes showed diminished spatiotemporal consistency in comparison to the baseline, while motion diversity was not significantly impacted. These results emphasize the importance of integrating SS into our pipeline, as corroborated by the UniCtrl's findings presented in Table 1, illustrating the essential role of SS in achieving the desired balance of spatiotemporal consistency and motion diversity.

Additionally, we conduct experiments on each individual component to provide a deeper understanding of their respective roles and interactions, as shown in Table 3. The results align with our previous analysis: when only SAC is present, we observe a significant increase in the DINO score, confirming the critical role of SAC in enhancing spatiotemporal consistency. In contrast, when only motion injection is applied without SAC and the consistency SAC brings, motion injection alone proves ineffective. Similarly, SS alone shows no impact; only when combined with SAC and MI does it effectively achieve spatiotemporal consistency.

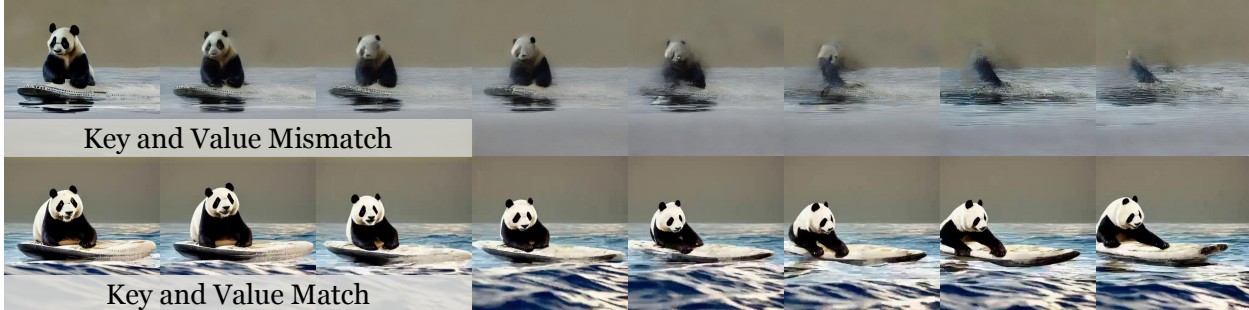

*A panda standing on a surfboard in the ocean under moonlight*

Figure 7: Each row displays a sequence of video frames generated from the identical prompt: "A panda standing on a surfboard in the ocean under moonlight." The section labeled **K and V mismatch** illustrates the frames produced when there is a discrepancy between key and value pairs. Conversely, the section titled **K and V match** showcases frames generated when key and value pairs are in alignment.

**The Impact of Motion Injection Degree**   To assess the impact of varying degrees of *motion injection*, we conducted experiments across both datasets using UniCtrl with *motion injection* degree set at $c = 0$, $c = 0.2$, $c = 0.4$, $c = 0.6$, $c = 0.8$, and $c = 1$. The effects of different *motion injection* levels are depicted quantitatively in Table 3. As the degree of *motion injection* escalates, we observed that the DINO score consistently outperforms baseline metrics, and the RAFT score progressively increases. This trend indicates an amplification in motion diversity. We showcase qualitative examples of the influence of *motion injection* degree in the supplementary material.

**The Impact of Swapping Key and Value**   We aim to provide a qualitative example to underscore the importance of simultaneously modifying both key and value, as highlighted in our discussion on the impact of key and value mismatches in Section 4.1. Initially, we alter only the value while maintaining the same key within the *cross-frame Self-Attention Control* (SAC) performing the UniCtrl pipeline. As depicted in the first row of Figure 7, the panda begins to fade in the subsequent frames, indicating a substantial decrease in the quality of the generated videos with respect to spatiotemporal consistency. This decline is particularly evident when compared to the approach of modifying both the key and value concurrently using the same UniCtrl pipeline.

## 6   Conclusion

We introduce UniCtrl to address the challenges of maintaining cross-frame consistency and preserving motions for Video Diffusion Models. By incorporating UniCtrl, we have notably improved the spatiotemporal consistency across frames of generated videos. Our approach stands out as it requires no additional training, making it adaptable to various underlying models. The efficacy of UniCtrl has been rigorously tested, demonstrating its potential to be widely applied to text-to-video generative models. We discuss the primary limitations of UniCtrl in Section 6.1 and detail our ethics statement in Section 6.2.

### 6.1   Limitations

Our method needs to operate on the attention mechanism, which limits the application of our method on non-attention-based models. Also, since we ensure the same value for each frame, changing colors within the video is not possible, which limits the model's ability to generate videos. Additionally, we have not yet guaranteed that spatial information is completely consistent across frames; this might be addressed in future work by controlling the temporal attention block. Furthermore, during the inference process, we need additional computation to preserve the motion query, which affects the inference speed. Our method can still be improved by addressing the above issues.

### 6.2   Ethics Statement

While UniCtrl offers significant advancements in video generation, it is imperative to consider its broader ethical, legal, and societal implications.

### 6.2.1 Copyright Infringement.

As an advanced video generation tool, UniCtrl could be utilized to modify and repurpose original video works, raising concerns over copyright infringement. It is crucial for users to respect the rights of content creators and uphold the integrity of the creative industry by adhering to copyright and licensing laws.

### 6.2.2 Deceptive Misuse.

Given its ability to generate high-quality, consistent video content, there is a risk that UniCtrl could be exploited for deceptive purposes, such as creating misleading or fraudulent content. This underscores the need for responsible usage guidelines and robust security measures to prevent such malicious applications and protect against security threats.

### 6.2.3 Bias and Fairness.

UniCtrl relies on underlying diffusion models that may harbor inherent biases, potentially leading to fairness issues in the generated content. Although our method is algorithmic and not directly trained on large-scale datasets, it is essential to acknowledge and address any biases present in these foundational models to ensure equitable and ethical utilization.

By proactively addressing these ethical considerations, we can responsibly leverage the capabilities of UniCtrl, ensuring its application aligns with legal standards and societal welfare. Emphasizing ethical practices, legal compliance, and the well-being of society is paramount in advancing video generation technology while maintaining public trust and upholding community values.

## Acknowledgment

We thank Raven, Tom, Martin, Yichi Zhang, Shengyi Qian for their helpful feedback and advice.

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

## Appendix

Here we present some of the sections left for discussion in the main paper. This supplementary report includes discussions on Details of Metrics, Implementation Details, and more Qualitative Results on the effectiveness of the motion injection degree, ablations of each module in UniCtrl, and how UniCtrl and FreeInit (Wu et al., 2023) can work together. These results further demonstrate UniCtrl's efficacy in improving spatiotemporal consistency and preserving motion dynamics.

## A  Details of Metrics

We consider standard metrics: DINO and RAFT following (Singer et al., 2022; Wu et al., 2023) and we present details of our evaluation metrics:

• *DINO*: To evaluate the spatiotemporal consistency in the generated video, we employ DINO (Oquab et al., 2024) to compare the cosine similarity between the initial frame and subsequent frames. The average DINO score across all consecutive frames is then used as the video's overall score. In our experiments, we utilize the DINO-vits16(Caron et al., 2021) model to compute the DINO cosine similarity. We present the details of our implementation in Algorithm 3.

• *RAFT*: To compare the magnitude of motion in the videos, we utilize RAFT (Teed & Deng, 2020) to estimate the optical flow, thereby inferring the degree of motion. To quantify the motion intensity between every two consecutive frames, we employ the RAFT model to estimate the optical flow for each pixel. We calculate the magnitude of all pixel optical flow using the $l^2$ norm, and finally, we obtain the average of all magnitude as the score for every consecutive frame in the video. In this process, we utilize the RAFT model from torchvision (maintainers & contributors, 2016). We present the details of the RAFT implementation in Algorithm 4.

---

**Algorithm 3** DINO Score
1: **Input:** Generated Video $v$
2: frames = video_to_frames($v$) # Obtain frames from generated video
3: features = DINO(frames) # Obtain features for each frame using DINO
4: **for** $n = 1$ to $N$ **do**
5:     scores = CosineSimilarity(features[0], features[$i$])
6: **end for**
7: $DINO\_SCORE = $ AVERAGE(scores)
8: **Output:** $DINO\_SCORE$

---

**Algorithm 4** RAFT Score
1: **Input:** Generated Video $v$
2: frames = video_to_frames($v$)
3: **for** $n = 1$ to $N$ **do**
4:     flow_vectors = RAFT(frames[$n-1$], frames[$n$])
5:     $scores = $ AVERAGE($\|$flow_vectors$\|$) # Calculate average flow magnitude
6: **end for**
7: $RAFT\_SCORE = $ AVERAGE(scores) # Average score from pairs of frames
8: **Output:** $RAFT\_SCORE$

---

## B  Implementation Details

### B.1  Backbones

Two open-sourced text-to-video models are used as the base models for FreeInit evaluation. For VideoCrafter(Chen et al., 2023), we adopt the VideoCrafter1-base-1024-T2V-model. For AnimateDiff (Guo

et al., 2023b), we use the mm-sd-v14 motion module with the Realistic-Vision-V5.1 LoRA model for evaluation. For AnimateLCM Wang et al. (2024), we use the AnimateLCM-sd15-t2v motion module and also Realistic-Vision-V5.1 LoRA model for evaluation.

### B.2 Inference Details

Experiments on VideoCrafter is conducted on $512 \times 320$ spatial scale and 16 frames, while experiments on AnimateDiff is conducted on a video size of $512 \times 512$, 16 frames. AnimateLCM is also conducted on 512 $\times$ 512, 16 frames. During the inference process, we use classifier-free guidance for all experiments including the comparisons and ablation studies, with a constant guidance weight 7.5. All experiments are conducted on Nvidia A10G GPU. We conduct experiments across different GPUs such as Nvidia A10G, Nvidia A40, and Nvidia A100 and we observe different inference results using the same seed while the CUDA (NVIDIA et al., 2020) random algorithm is deterministic. We resolve this behavior by initializing latent using NumPy (Harris et al., 2020). We hope this change will further enhance the reproducibility of our code across different CUDA architectures.

## C Detailed Quantitative Results Tables

Since our experiments are conducted with multiple seeds, we report the mean values and standard deviations in the following table as a supplement to the previous tables

Table 4: Detailed Table for Table 1

| Method | DINO ($\uparrow$) | RAFT ($\uparrow$) |
|---|---|---|
| AnimateDiff (Guo et al., 2023b) | $94.26 \pm 0.63$ | $25.44 \pm 5.81$ |
| FreeInit + AnimateDiff ($I = 3$) | $96.04 \pm 0.22$ | $11.62 \pm 1.72$ |
| UniCtrl + AnimateDiff ($c = 1$) | $96.38 \pm 0.21$ | $23.29 \pm 2.09$ |
| VideoCrafter (Chen et al., 2023) | $93.53 \pm 0.27$ | $29.20 \pm 1.16$ |
| FreeInit + VideoCrafter ($I = 3$) | $96.75 \pm 0.07$ | $7.46 \pm 0.59$ |
| UniCtrl + VideoCrafter ($c = 1$) | $95.55 \pm 0.17$ | $25.11 \pm 1.98$ |
| AnimateLCM (Wang et al., 2024) | $92.96 \pm 0.18$ | $20.80 \pm 5.15$ |
| FreeInit + AnimateLCM ($I = 3$) | $94.90 \pm 0.89$ | $14.64 \pm 4.59$ |
| UniCtrl + AnimateLCM ($c = 1$) | $95.40 \pm 0.27$ | $17.53 \pm 3.47$ |
| UniCtrl + FreeInit + AnimateDiff ($I = 3, c = 1$) | $96.85 \pm 0.19$ | $9.82 \pm 1.13$ |

Table 5: Detailed Table for Table 2

| Method | FVD ($\downarrow$) | FVMD ($\downarrow$) |
|---|---|---|
| AnimateDiff (Guo et al., 2023b) | $1069.90 \pm 57.89$ | $27124.17 \pm 3234.76$ |
| FreeInit + AnimateDiff | $958.97 \pm 40.82$ | $25078.05 \pm 5490.52$ |
| UniCtrl + AnimateDiff | $819.74 \pm 29.42$ | $8864.07 \pm 577.43$ |

Table 6: Detailed Table for Table 3

| Method | DINO ($\uparrow$) | RAFT ($\uparrow$) |
|---|---|---|
| AnimateDiff (Guo et al., 2023b) | $94.26 \pm 0.63$ | $25.44 \pm 5.81$ |
| UniCtrl w/o SAC + AnimateDiff | $94.26 \pm 0.62$ | $25.42 \pm 5.86$ |
| UniCtrl w/o MI + AnimateDiff | $98.08 \pm 0.47$ | $4.12 \pm 2.33$ |
| UniCtrl w/o SS + AnimateDiff | $94.26 \pm 1.16$ | $21.90 \pm 2.08$ |
| only SAC + AnimateDiff | $98.08 \pm 0.47$ | $4.12 \pm 2.33$ |
| only MI + AnimateDiff | $94.26 \pm 0.62$ | $25.42 \pm 5.86$ |
| only SS + AnimateDiff | $94.26 \pm 0.62$ | $25.42 \pm 5.86$ |
| UniCtrl ($c = 0$) + AnimateDiff | $98.08 \pm 0.47$ | $4.12 \pm 2.33$ |
| UniCtrl ($c = 0.2$) + AnimateDiff | $97.41 \pm 0.10$ | $9.04 \pm 1.37$ |
| UniCtrl ($c = 0.4$) + AnimateDiff | $96.69 \pm 0.14$ | $15.56 \pm 1.44$ |
| UniCtrl ($c = 0.6$) + AnimateDiff | $96.46 \pm 0.17$ | $20.16 \pm 1.73$ |
| UniCtrl ($c = 0.8$) + AnimateDiff | $96.37 \pm 0.19$ | $22.50 \pm 1.90$ |
| UniCtrl ($c = 1.0$) + AnimateDiff | $96.38 \pm 0.21$ | $23.29 \pm 2.09$ |

## D    More Qualitative Results

### D.1    Motion Injection Degree

In the main paper, we quantitatively demonstrate that higher degrees of motion injection results in videos with more pronounced motion. Here, we provide a qualitative example in Fig. 8. The motion of the Corgi is significantly more pronounced when comparing videos generated with a higher motion injection degree to those with a lower degree, while the appearance of the Corgi remains consistent across frames in each video.

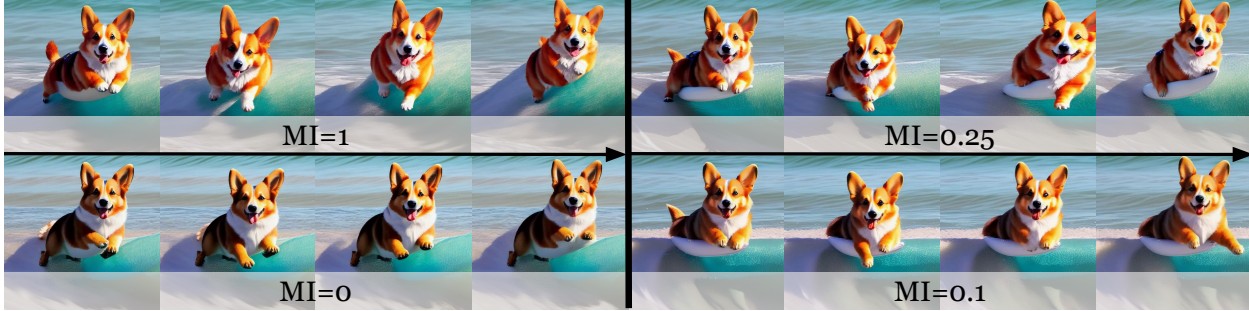

*A Corgi is surfing*

Figure 8: We present 4 sequences of images inferenced from the same prompt but with different Motion Injection Degree. With larger motion injection degree, corgi in each generated video presents more motion while preserve its appearance.

### D.2    Qualitative Ablation Results

In the main paper, we conduct quantitative ablation studies of each module and we would like to emphasize the indispensable nature of every component within UniCtrl again by presenting Fig. 9. This figure qualitatively illustrates the ablations through a single set of results. By comparing the first and second rows, it becomes evident that the results in the second-row display inconsistencies across frames, both in terms of the car and the road. Intuitively, the motion branch and output branch keep deviating while denoising. SS effectively bridges the difference between the motion branch and the output branch and significantly improves spatiotemporal consistency as a result. By comparing the first and third rows, we observe the third row's result shows much more inconsistencies which clearly explains the necessity of incorporating SAC in UniCtrl. Lastly, we show the fourth row's result contains little motion compared with the first row's result. Thus, we prove that MI is also one of the indispensable modules of UniCtrl.

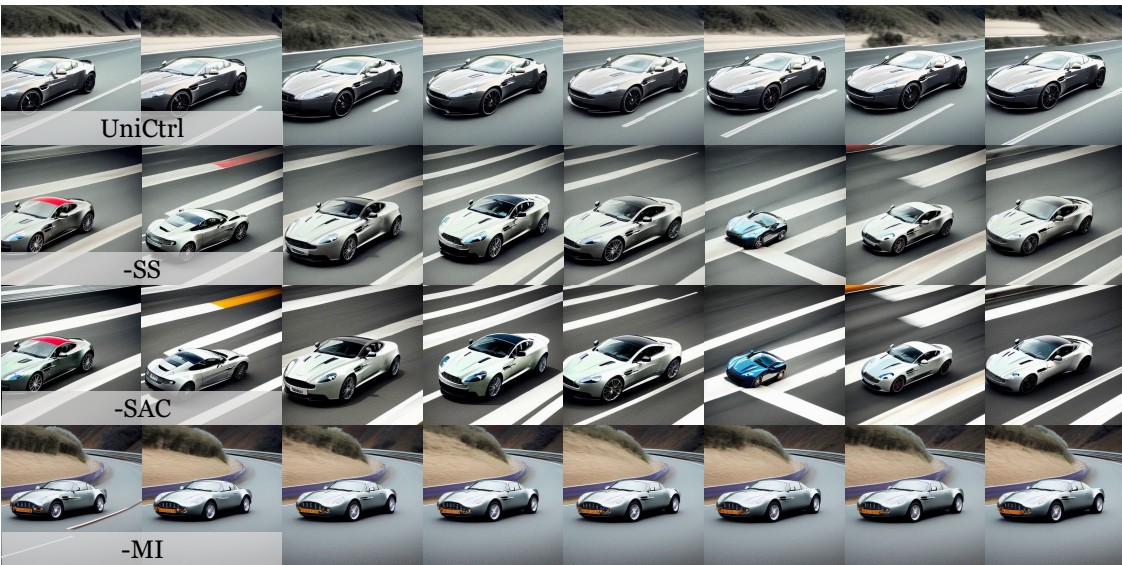

*Aston Martin is moving*

Figure 9: The first row demonstrates results generated with UniCtrl, which the vehicle and the road are both consistent across the frames. The second row shows frames generated with SAC and MI. The third row explains frames augmented with SS and MI. The fourth row contains frames shows results with SS and SAC. These comparisons serve as qualitative examples for ablation for each module in UniCtrl.

### D.3 Unification between UniCtrl and FreeInit

We state that UniCtrl and FreeInit explore orthogonal directions to improve the spatiotemporal consistency of video diffusion models. Now we want to first explain how to unify UniCtrl and FreeInit. In the framework of FreeInit, it requires $N$ FreeInit iterations, $N = 3$ in our implementation, and we apply UniCtrl during the first iteration. We find the unification between UniCtrl and FreeInit improves the quality of generated videos and we showcase additional qualitative examples in Fig. 10.

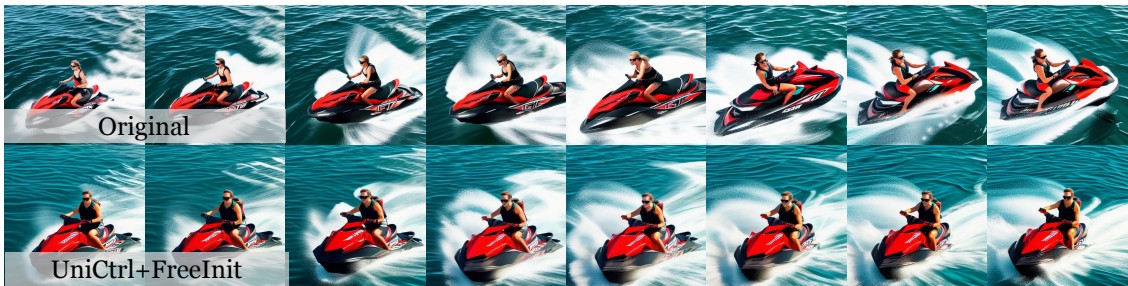

*jet ski on the water*

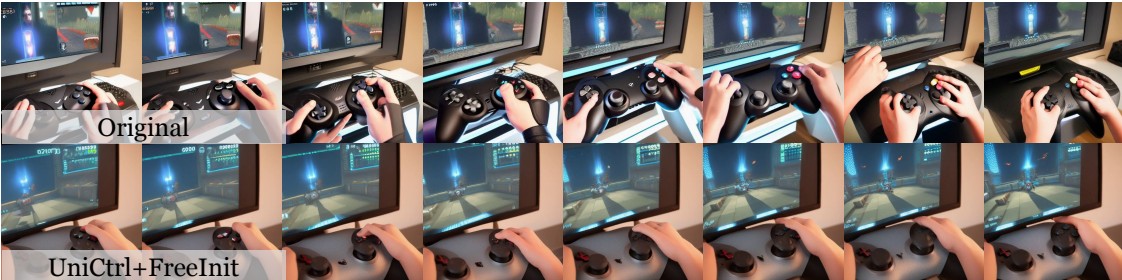

*person plays a video game*

Figure 10: We present more qualitative examples in order to show the unification between UniCtrl and FreeInit can improve spatiotemporary consistency and preserve motion dynamics, which further demonstrate that UniCtrl and FreeInit are orthogonal works that both can improve spatiotemporary consistency.

