# OpenReview forum: "UniCtrl: Improving the Spatiotemporal Consistency of Text-to-Video Diffusion Models via Training-Free Unified Attention Control"
_TMLR — Accepted by TMLR_

### Review · Reviewer_hjXg · 2024-10-05

**Summary Of Contributions:**

This paper introduces a systematic framework for enhancing the spatiotemporal consistency of text-to-video diffusion models. The proposed method, UniCtrl, leverages cross-frame self attention control to maintain spatial consistency, motion injection to preserve motion magnitude and spatiotemporal synchronization to further enhance consistency. The proposed method is tested on pretrained T2V backbones such as AnimateDiff and VideoCrafter and showed improved results on metrics such as DINO and RAFT.

**Audience:**

Yes

**Claims And Evidence:**

No

**Requested Changes:**

See weaknesses

**Strengths And Weaknesses:**

Strengths:
1. The proposed method is a plug-and-play module that does not require any finetuning.
2. The method shows enhanced spatial consistency of the objects in the video compared to baseline methods.

Weaknesses:
1. While the cross-frame self-attention control module improves spatial consistency and reduces the chance of object appearance changing in the generated video, it also heavily affects the motion magnitude of the video. Based on the examples the author shared, even after applying motion injection and spatiotemporal injection, the generated video with UniCtrl still suffers from low motion magnitude or near static video scenes. I'm wondering if there is a way to control the strength of UniCtrl affecting the video content such that it can strike a balance between consistency and motion degree. Can the author provide a more detailed analysis regarding this issue?
2. The proposed motion injection is based on cross-attention control. It is interesting that the author did not consider the temporal attention layers, which are often considered as the key module for modelling temporal information in T2V models. Can the author provide more insights on the design considerations of the motion injection mechanism?
3. The experiments only considered two metrics and did not include any video quality metrics such as FVD and video inception score (see appendix G in [1]). It is unknown whether UniCtrl generates better quality videos compared to baseline methods and FreeInit. Also, since the method is designed to enhance text-to-video generation, it would be interesting if the author could provide more analysis on video-text metrics such as CLIPSIM, to indicate if there are differences in the alignment with text prompts before and after applying UniCtrl.
4. The paper could benefit from including qualitative examples with more diverse frames and higher motion magnitudes. For example, currently in all 4 examples in Figure 2, it is hard to recognize whether the car is actually moving forward or not.
5. While the paper claims that UniCtrl can be applied to any T2V models, only AnimateDiff examples are shown in the paper and there is no VideoCrafter1 examples. From the experimental results, UniCtrl + VideoCrafter1 also suffers from a decreased DINO score compared to FreeInit. The author may want to provide more qualitative and quantitative results to support their claim on the universality of UniCtrl.

[1] Blattmann, Andreas, et al. "Align your latents: High-resolution video synthesis with latent diffusion models." Proceedings of the IEEE/CVF Conference on Computer Vision and Pattern Recognition. 2023.

---

> ### Author Response · Authors · 2024-10-17
> **Response to Reviewer hjXg**
>
> We sincerely appreciate your insightful feedback on our work. In this response, we aim to address your concerns one by one. We have also uploaded a revised version of the paper that incorporates your suggestions.
>
> **Regarding Weakness 1:**
> We would like to mention that we provide a motion injection degree c, to control the strength of the motion, allowing for a balance between consistency and motion degree. As demonstrated in the ablation study (Table 3), varying c from 0 to 1 shows a decrease in DINO and an increase in RAFT, confirming that c effectively regulates the motion magnitude and achieves the desired trade-off.
> | **Method**                               | DINO (↑) | RAFT (↑) |
> |------------------------------------------|--------------|--------------|
> | AnimateDiff   | 94.26    | 25.44    |
> | UniCtrl (c = 0) + AnimateDiff            | 98.08        | 4.12         |
> | UniCtrl (c = 0.2) + AnimateDiff          | 97.41        | 9.04         |
> | UniCtrl (c = 0.4) + AnimateDiff          | 96.69        | 15.56        |
> | UniCtrl (c = 0.6) + AnimateDiff          | 96.46        | 20.16        |
> | UniCtrl (c = 0.8) + AnimateDiff          | 96.37        | 22.50        |
> | UniCtrl (c = 1.0) + AnimateDiff          | 96.38        | 23.29        |
>
>
> **Regarding Weakness 2:**
> As we understand it, the purpose of temporal attention layers is to enhance the smoothness of frames and improve temporal consistency. Modifying them to achieve the opposite effect would be kind counterintuitive. Therefore, we chose to focus on changing spatial attention layers, leaving further exploration of temporal attention for future work.
>
> **Regarding Weakness 3:**
> We have added experiments for FVD and FVMD in Table 2 of our revised version to better evaluate both video quality and motion quality. Regarding CLIPSIM, we conducted experiments for the original and UniCtrl respectively.
> | CLIPSIM                  | seed = 42   | seed = 17   | seed = 0    | Average     | std         |
> |--------------------------|-------------|-------------|-------------|-------------|-------------|
> | AnimateDiff orig.         | 28.65101068 | 28.90661987 | 28.53958092 | 28.69907049 | 0.1881799851 |
> | AnimateDiff Unictrl       | 28.63654499 | 28.78042822 | 28.60973694 | 28.67557005 | 0.09179375923 |
>
>  The difference is minimal, which is expected since we did not significantly alter the behavior of cross-attention for text-prompt.
>
> **Regarding Weakness 4:**
> Thanks for your feedback. We acknowledge that it is challenging to distinguish the motion of those 4 exmaples effectively through still images in the PDF. In cases of significant motion in the PDF, the entire video may become heavily crashed, as we typically do not expect substantial motion change within a span of 16 frames. However, we provided video examples in the supplementary materials, which could provide a clearer demonstration of the motion. We hope this could addresses the concern.
>
> **Regarding Weakness 5:**
> We have included examples of Videocrafter and AnimateLCM in the Figure 5 of revised version, as well as additional experiments on the AnimateLCM model in experiments section to support our claims. We observe a significant improvement in the DINO scores across all three models compared to the original and high RAFT scores, indicating enhanced spatiotemporal consistency while preserving high-quality motion.

---

> > ### Comment · Reviewer_hjXg · 2024-10-27
> >
> > Thanks for the reply. Most of my concerns have been addressed. I'd suggest choosing better examples to replace the "rabbit" and "corgi" examples in Figure 5 as the generated results for UniCtrl seem to be completely static.

---

> > > ### Author Response · Authors · 2024-10-29
> > > **Thank you!**
> > >
> > > Thank you for your valuable suggestions! In the updated version, we have changed those two examples in Figure 5.

---

### Review · Reviewer_SM4o · 2024-10-05

**Summary Of Contributions:**

This paper proposes a couple of useful techniques to improve the spatiotemporal consistency of pre-trained text-to-video diffusion models without the need for GPU resources to fine-tune the model. The core of the approach lies in the smart use and rearrangement of key components in the attention layer. The key methodologies involve: 1) cross-frame self-attention control, 2) motion injection, and 3) spatiotemporal synchronization.

**Audience:**

Yes

**Claims And Evidence:**

Yes

**Requested Changes:**

This paper focuses heavily on engineering implementation and empirical performance rather than theoretical aspects. The proposed methods are simple and appear effective in addressing some of the problems in current video generation pipelines. However, to make the contributions more solid and the claims clearly supported, I would like the authors to address the above concerns (mostly related to the experimental side) to improve the paper's quality.

**Strengths And Weaknesses:**

Strengths:
1. The writing is clear and easy to follow. The video samples in the supplementary materials are straightforward to compare, effectively validating the proposed method's efficacy.
2. The proposed methods are simple and easy to implement for video diffusion models (VDMs) using attention modules.

Weaknesses:
1. It would be beneficial to provide more quantitative results using dedicated metrics for video motion consistency, such as FVMD [1], especially to highlight the effectiveness of the motion injection module. I don't think RAFT alone is sufficient, as a larger magnitude of motion (which can be captured by RAFT) does not necessarily indicate better video quality. A poor video generative model may produce inconsistent motion and still result in a large amount of motion/RAFT value. Simply put, a larger amount of motion is not naturally equivalent to better video quality or diversity. The evaluation of the motion injection module needs to take this into account.

2. More ablation studies on the effect of the motion injection parameter $c$ are needed. Currently, the early denoising stages use motion queries in the output branch to enhance motion diversity, while the later denoising steps do not employ this technique. The cutoff is applied at a single timestep. Given the importance of this parameter, as highlighted in the experimental sections, a more detailed discussion on the design of the switch schedule would be valuable. For instance, does implementing a linear interpolation (between two queries) over a range of transition timesteps instead of a single cutoff work? Additional engineering details would assist researchers in the community to more easily adapt this method.

3. In the introduction, it is better to emphasize that attention modules are widely used and explain why they are crucial components in modern text-to-video generation pipelines. It seems somewhat taken for granted that VDMs heavily rely on attention modules from the text.

4. Some recent works on video-to-video editing share similar features, particularly the focus on spatiotemporal consistency, but are not mentioned in this paper (for example, [2][3]). The authors are encouraged to refer to recent surveys, such as [4], to discuss related works in this field with a similar focus. Although the specific methods differ, the common goal of enhancing spatiotemporal and motion consistency is evident in recent studies.


References:  \
[1] Fréchet Video Motion Distance: A Metric for Evaluating Motion Consistency in Videos, ICML 2024 CVG Workshop. \
[2] Style-a-video: Agile diffusion for arbitrary text-based video style transfer, IEEE Signal Processing Letters (2024). \
[3] Rerender a video: Zero-shot text-guided video-to-video translation. SIGGRAPH Asia 2023 Conference Papers. 2023. \
[4] Video Diffusion Models: A Survey, preprint.

---

> ### Author Response · Authors · 2024-10-17
> **Response to Reviewer SM4o**
>
> We greatly appreciate your valuable feedback and would like to address your concerns point by point. We have also uploaded a revised version of the paper that incorporates your suggestions.
>
> **Regarding Weakness 1:**
> Thank you for your constructive input. We have added experiments for FVD and FVMD[1] in our revised version to better evaluate both video quality and motion quality.
>
> **Regarding Weakness 2:**
>
> This is an interesting point to explore—whether to use linear interpolation or a simple cutoff or even other interpolations. However, since our main focus remains on spatiotemporal consistency, we do not believe we can address this question thoroughly at this stage. We hope to explore it in future work.
>
> **Regarding Weakness 3:**
>
> Thanks for the advise. We have edited the introduction in the paper.
>
> **Regarding Weakness 4:**
>
> Thank you for your feedback. We hope this work will encourage more researchers in the field to focus on the issue of spatiotemporal consistency. We have also included [2,3,4] as a reference.

---

> > ### Comment · Reviewer_SM4o · 2024-10-22
> >
> > Thank you for your efforts in addressing the technical concerns. I don't have any further questions at this time.

---

> > > ### Author Response · Authors · 2024-10-23
> > > **Thank you!**
> > >
> > > Thank you for your valuable feedback!

---

### Review · Reviewer_ykrr · 2024-10-07

**Summary Of Contributions:**

This paper is motivated by the problem that the Diffusion video generation framework frames are not in accordance with the other frames in the video. It presents 3 submethods that work on any transformer-based video generation model, namely, SAC, MI, and SS.

**Audience:**

Yes

**Broader Impact Concerns:**

Have been addressed in text

**Claims And Evidence:**

No

**Requested Changes:**

1) I would like to see UniCtrl compared with MI only. Given the ablation numbers, I suspect that UniCtrl can be reduced to Motion Injection (MI).
2) All experiments done with multiple seeds and the standard error should be reported.

**Strengths And Weaknesses:**

Strength:
1) The ablation study is extensive from the number of setups point of view.
2) The writing is clear and easy to follow.
3) MI seems to be a promising module which enhances the motion in the generated video while decreasing the consistency between frames, which is understandable.
4) The limitation have been included.

Weakness:

1) [Major] Given the stochastic nature of diffusion models the experiment section should image multiple runs with different random seeds and also report the standard deviation in the score.
2) [Major] From the ablation study MI seems to be the most impactful module and other modules seem to have little to no effect. Given the results of the ablation study it is hard to conclude the usefulness of SAC and SS. Given this the writing claim with respect to SAC and SS need to be significantly revised. There is a clear mismatch between claims and evidence.
3) [Minor] More methods need to be compared in the baseline.

---

> ### Author Response · Authors · 2024-10-17
> **Response to Reviewer ykrr**
>
> We sincerely appreciate your valuable feedback and aim to address your concerns one by one. We have also uploaded a revised version of the paper that incorporates your suggestions.
>
> **Regarding Weakness 1:**
> Thanks for your advice. We have conducted all experiments in multiple seeds, and report the average score in the main paper and corresponding standard error in Appendix C Detailed Quantitative Results Tables.
>
> **Regarding Weakness 2:**
> We have also conducted expirements for every compenents independently alongside the ablation study：
> | Method                          | DINO (↑) | RAFT (↑) |
> |----------------------------------|----------|----------|
> | AnimateDiff   | 94.26    | 25.44    |
> | UniCtrl w/o SAC + AnimateDiff    | 94.26    | 25.42    |
> | UniCtrl w/o MI + AnimateDiff     | 98.08    | 4.12     |
> | UniCtrl w/o SS + AnimateDiff     | 94.26    | 21.90    |
> | only SAC + AnimateDiff           | 98.08    | 4.12     |
> | only MI + AnimateDiff            | 94.26    | 25.42    |
> | only SS + AnimateDiff            | 94.26    | 25.42    |
>
> It is unsurprising that the score for 'only MI + AnimateDiff' is similar to the original. MI is a method designed to compensate for low motion magnitude, but it does not offer improvements on its own. In contrast, SAC plays a crucial role in enhancing spatial consistency, which is why it can achieve the highest DINO score independently. However, to address the issue of low motion, we introduced both MI and SS. SS is also essential for maintaining spatial consistency when MI is enabled. Without SS, the results degrade to the level of the original in terms of the DINO score, showing no improvement in spatial consistency. Therefore, the presence of both SAC and SS is vital, by providing spatiotemporal consistency, while MI is designed for only motion injection.
>
> **Regarding Weakness 3:**
> To the best of our knowledge, training-free methods that improve spatiotemporal consistency are relatively scarce. While we could not identify additional suitable baselines for comparison, we are open to including more if have.

---

> > ### Comment · Reviewer_ykrr · 2024-11-05
> > **Abalation for VideoCrafter**
> >
> > Thank you for the complete ablation and explanation of the components used. Will it be possible to also present ablation for **VideoCrafter** so that we may observe the same effects? That will make the fact that these modules together are method agnostics and make the performance better in a similar way.

---

> ### Author Response · Authors · 2024-11-06
> **Response to Abalation for VideoCrafter**
>
> Thanks for the valuable advice.
>
> We conducted ablation experiments for VideoCrafter:
>
> | Method                          | DINO (↑) | RAFT (↑) |
> |----------------------------------|----------|----------|
> | VideoCrafter   | 93.53    | 29.20    |
> | UniCtrl + VideoCrafter   | 95.55    | 25.11    |
> | UniCtrl w/o SAC + VideoCrafter    | 93.62   | 27.99 |
> | UniCtrl w/o MI + VideoCrafter     | 98.83  | 0.44     |
> | UniCtrl w/o SS + VideoCrafter     | 94.09    | 27.11    |
> | only SAC + VideoCrafter           | 98.83    | 0.44   |
> | only MI + VideoCrafter            | 93.62    | 27.99    |
> | only SS + VideoCrafter            | 93.62    | 27.99    |
>
> The pattern observed here closely mirrors that in AnimateDiff. Dropping SAC significantly reduces spatial consistency, resulting in a substantial decrease in the DINO score. Without MI, motion becomes almost negligible, leading to a static video with an extremely small RAFT score. Additionally, removing SS greatly impacts spatial consistency, and the DINO score also decreases significantly. These findings allow us to draw the same conclusions to those of AnimateDiff. Moreover, each independent component behaves identically to those in AnimateDiff, which aligns with expectations.

---

### Review · Reviewer_ndyH · 2024-10-08

**Summary Of Contributions:**

This paper proposes a training-free method to enhance the spatiotemporal consistency and motion diversity of videos generated by existing text-to-video models. The approach incorporates cross-frame self-attention control to ensure semantic consistency across frames, and motion injection to improve motion quality. Experiments are conducted using widely-used text-to-video models, such as AnimateDiff and VideoCrafter, to verify the method’s effectiveness.

**Audience:**

Yes

**Claims And Evidence:**

Yes

**Requested Changes:**

Please see the weakness part.

**Strengths And Weaknesses:**

Strengths:
* The paper is well-written and the motivation is straightforward.
* The proposed method is training-free and can be employed in various attention-based video diffusion models.

Weakness:
* My largest concern is that the proposed attention control method somewhat mitigates temporal inconsistency, but it may result in limited or near-static motion. This trade-off suggests that while the method enhances stability, it may reduce the diversity and realism of movement, especially in dynamic scenes.
* The experiments only report DINO and RAFT metrics, omitting widely-used metrics such as FVD, KVD, and CLIP in the evaluation.
* All visual examples are based on the AnimateDiff model, with no results presented for VideoCrafter.
* The improvement of the proposed method appears to be limited and trade-offs. According to Table 1, the DINO score is lower than that of FreeInit+baseline. The improvement is seen only in the RAFT metric, possibly due to the generation of “small motion” results. Therefore, further evaluation using additional metrics is necessary to fully verify its effectiveness.
* The proposed method is only compared to FreeInit. Additional comparisons with other competing methods are needed.
* The paper only selected a few text-to-video models as baselines. Since the proposed method relies on K and V features from the first frame, it may be more suited for image-to-video models. Additional experiments are needed to further support the claims.

---

> ### Author Response · Authors · 2024-10-17
> **Response to Reviewer ndyH**
>
> We greatly appreciate your valuable feedback and would like to address your concerns point by point. We have also uploaded a revised version of the paper that incorporates your suggestions.
>
> **Regarding Weakness 1:**
>
> Thanks for your suggestions. This is the primary reason on why we are proposing motion injection. We acknowledge that in our approach, methods designed to improve spatiotemporal consistency can limit the diversity of motion—this is unavoidable. However, compared to previous work such as Freeinit, which only focus on temporal inconsistency without motion, we introduce motion injection to mitigate this issue. From our experiments, we found that our motion injection works effectively, bringing RAFT performance close to the original level when the motion injection degree is set to 1.
> | **Method**                               | DINO (↑) | RAFT (↑) |
> |------------------------------------------|--------------|--------------|
> | AnimateDiff   | 94.26    | 25.44    |
> | UniCtrl (c = 0) + AnimateDiff            | 98.08        | 4.12         |
> | UniCtrl (c = 1.0) + AnimateDiff          | 96.38        | 23.29        |
>
> **Regarding Weakness 2:**
>
> We have added experiments for FVD and FVMD in our revised version to better evaluate both video quality and motion quality.
>
> We also gathered results for CLIP:
>
> | CLIPSIM                  | seed = 42   | seed = 17   | seed = 0    | Average     | std         |
> |--------------------------|-------------|-------------|-------------|-------------|-------------|
> | AnimateDiff orig.         | 28.65101068 | 28.90661987 | 28.53958092 | 28.69907049 | 0.1881799851 |
> | AnimateDiff Unictrl       | 28.63654499 | 28.78042822 | 28.60973694 | 28.67557005 | 0.09179375923 |
>
> which shows no significant difference.
>
> **Regarding Weakness 3:**
>
> We have included qualitative results for VideoCrafter and AnimateLCM in the Figure 5 of the updated version paper.
>
> **Regarding Weakness 4:**
>
> We have included more results and udpated the table in order to support our claim. Here is an abbreviated table to showcase that our method helps the video diffusion model to generate content with improved quality and motion consistency:
> | Method                         | FVD (↓)  | FVMD (↓)   |
> |---------------------------------|----------|------------|
> | AnimateDiff                     | 1069.90  | 27124.17   |
> | FreeInit + AnimateDiff          | 958.97   | 25078.05   |
> | UniCtrl + AnimateDiff           | 819.74   | 8864.07    |
>
> **Regarding Weakness 5:**
>
> To the best of our knowledge, training-free methods that improve spatiotemporal consistency are relatively scarce. While we could not identify additional suitable baselines for comparison, we are open to including more if have.
>
> **Regarding Weakness 6:**
>
> We have added AnimateLCM as another backbone in the updated version. We acknowledge that some literature has touched on this insight. However, we currently do not plan to evaluate image-to-video models as it falls outside the primary scope of our work.

---

> > ### Comment · Reviewer_ndyH · 2024-10-29
> > **Thank you for the response.**
> >
> > Thank you for your response. I still have one concern regarding the evaluation of FVD. Using only 100 unique prompts seems too few to effectively evaluate the FVD, e.g., FreeInit uses 2048 samples to caculate it.

---

> > > ### Author Response · Authors · 2024-10-29
> > > **Thank you!**
> > >
> > > Thank you for your helpful suggestions!
> > >
> > > Regarding the remaining concern, we would like to clarify how Freeinit outlines its approach. In section 5.1, Freeinit states, “For UCF-101, we use the same prompt list as proposed in [1],” and in their FVD experiment, “We follow [1] to perform zero-shot text-to-video generation on UCF-101 and sample 2,048 videos to compute the FVD.” The prompt list here is a list of only 105 unique prompts, which is used in both the Freeinit evaluation and ours. We follow the experiments in Freeinit and [1] exactly, using these prompts to generate and sample 2,048 videos to compute the score.
> > >
> > > [1] Preserve Your Own Correlation: A Noise Prior for Video Diffusion Models

---

> > > > ### Comment · Reviewer_ndyH · 2024-11-04
> > > >
> > > > Thanks for your reply. I do not have further concerns.

---

### Review · Reviewer_63d9 · 2024-10-10

**Summary Of Contributions:**

This work presents a plug-and-play method to enhance spatiotemporal consistency and motion diversity in T2V generation. It comprises three components: cross-frame self-attention control, motion injection, and spatiotemporal synchronization. The method requires no additional training and is applicable to various T2V models. Extensive experiments demonstrate its effectiveness.

**Audience:**

Yes

**Claims And Evidence:**

Yes

**Requested Changes:**

See weaknesses.

**Strengths And Weaknesses:**

Strengths:
1. The paper is well-written.
2. The motivations of the proposed components are intuitive and reasonable.
3. The evaluation and ablation study are convincing.

Weaknesses:
1. To my understanding, in motion injection, when the input latent is copied, the Qs of self-attention in the output branch and the motion branch should be the same. If so, the only operation would be the Q replacement in cross attention. The authors should clarify this point.
2. From Table 1, integrating both UniCtrl and FreeInit into AnimateDiff yields lower RAFT compared to only using UniCtrl. Can the authors provide the intuition behind this observation?
3. In Table 2, the authors only report the results of the method without each component. I would like to see the results of the method with each component, which may better show each component’s effect.
4. I believe that substantial constraints, such as using KV from the first frame and copying the latent, would significantly limit the diversity of the generated videos, even though motion injection may alleviate this issue.

---

> ### Author Response · Authors · 2024-10-17
> **Response to Reviewer 63d9**
>
> We greatly appreciate your valuable feedback and would like to address your concerns point by point. We have also uploaded a revised version of the paper that incorporates your suggestions.
>
> **Regarding Weakness 1:**
>
> Thanks for your advice. We have clarified this point in Figure 3 of updated version.
>
> **Regarding Weakness 2:**
>
> Given that we are also utilizing the Freeinit iterations, which only improve spatial consistency and decrease the motion, the motion of the video is expected to decrease compared to Unictrl when Freeinit is added.
>
> **Regarding Weakness 3:**
>
> Thanks for your suggestions. We have added experiments for each component individually in our updated version and discussed them in the ablation study.
>
> **Regarding Weakness 4:**
>
> Thanks for your advice. We acknowledge that in our approach, methods designed to improve spatiotemporal consistency can limit the diversity of motion—this is unavoidable. However, compared to previous work such as Freeinit, which only focus on inconsistency without motion, we introduce motion injection to mitigate this issue. From our experiments, we found that our motion injection works effectively, bringing RAFT performance close to the original level when the motion injection degree is set to 1.
> | **Method**                               | DINO (↑) | RAFT (↑) |
> |------------------------------------------|--------------|--------------|
> | AnimateDiff   | 94.26    | 25.44    |
> | UniCtrl (c = 0) + AnimateDiff            | 98.08        | 4.12         |
> | UniCtrl (c = 1.0) + AnimateDiff          | 96.38        | 23.29        |

---

> > ### Comment · Reviewer_63d9 · 2024-10-22
> > **Official Comment by Reviewer 63d9**
> >
> > Thanks for the reply. My concerns are mostly addressed.

---

> > > ### Author Response · Authors · 2024-10-23
> > > **Thank you!**
> > >
> > > Thank you for your helpful feedback!

---

### Decision · Action_Editor_gyz5 · 2024-11-06

**Recommendation:** Accept with minor revision

**Comment:**

In summary, the paper is well-executed and provides a meaningful contribution in video generation refinement. All reviewers are leaning towards acceptance. However, the reviewers suggest selecting better examples to better showcase the results. Thus, minor revisions are required, primarily to improve the presentation of qualitative results.

**Audience:**

The researches who focus on text-to-video generation and training, fine-tuning, or adapting diffusion models will be interested in the findings of this work.

**Claims And Evidence:**

The claims of this work are supported by the experiments of the comparison results and ablation study.